# Ccq1–Raf2 interaction mediates CLRC recruitment to establish heterochromatin at telomeres

Shaohua Shi[1,2,*], Yuanze Zhou[4,*], Yanjia Lu[1,2], Hong Sun[5,6], Jing Xue[5], Zhenfang Wu[1,2], Ming Lei[1,2,3]

**Telomeres, highly ordered DNA-protein complexes at eukaryotic linear chromosome ends, are specialized heterochromatin loci conserved among eukaryotes. In *Schizosaccharomyces pombe*, the shelterin complex is important for subtelomeric heterochromatin establishment. Despite shelterin has been demonstrated to mediate the recruitment of the Snf2/histone deacetylase–containing repressor complex (SHREC) and the Clr4 methyltransferase complex (CLRC) to telomeres, the mechanism involved in telomeric heterochromatin assembly remains elusive due to the multiple functions of the shelterin complex. Here, we found that CLRC plays a dominant role in heterochromatin establishment at telomeres. In addition, we identified a series of amino acids in the shelterin subunit Ccq1 that are important for the specific interaction between Ccq1 and the CLRC subunit Raf2. Finally, we demonstrated that the Ccq1–Raf2 interaction is essential for the recruitment of CLRC to telomeres, that contributes to histone H3 lysine 9 methylation, nucleosome stability and the shelterin-chromatin association, promoting a positive feedback mechanism for the nucleation and spreading of heterochromatin at subtelomeres. Together, our findings provide a mechanistic understanding of subtelomeric heterochromatin assembly by shelterin-dependent CLRC recruitment to chromosomal ends.**

## Introduction

Eukaryotic genomes are organized into structurally and functionally distinct domains, with euchromatin and heterochromatin transcriptionally active and repressed, respectively (Jenuwein & Allis, 2001). The chromosomal regions that contain large amounts of repetitive DNA elements are the main loci for heterochromatin formation (Richards & Elgin, 2002; Grewal & Jia, 2007). In fission yeast *Schizosaccharomyces pombe*, heterochromatin is mainly present at three regions, centromeres, silent mating-type loci and telomeres (Grewal, 2000). Extensive studies on heterochromatin assembly at centromeres and silent mating-type loci have revealed RNAi- and DNA-mediated pathways of heterochromatin nucleation in *S. pombe* (Hall et al, 2002; Volpe et al, 2002; Jia et al, 2004). The RNAi pathway, which is well characterized at centromeres, involves Dcr1-mediated siRNA production, histone posttranslational modifications by histone deacetylases (HDACs) and histone methyltransferases (HMTs), and recruitment of heterochromatin protein Swi6 (a homolog of mammalian HP1 protein) (Nakayama et al, 2001; Motamedi et al, 2004; Verdel et al, 2004; Kato et al, 2005; Sugiyama et al, 2005; Bühler et al, 2006). Heterochromatin nucleation by DNA-binding factors was revealed by the observation that Atf1/Pcr1 functions in parallel with RNAi pathway to nucleate heterochromatin at silent mating-type loci (Jia et al, 2004). In contrast to the well-characterized mechanisms of heterochromatin assembly at centromeres and silent mating-type loci, the nucleation and maintenance of heterochromatin at telomeres are still not fully understood.

Telomeres are specialized heterochromatin at eukaryotic linear chromosome ends that contribute to genome integrity (Tadeo et al, 2013). Like most eukaryotes, the *S. pombe* telomeric DNA consists of a track of double-stranded G-rich repeats and a short protruding single-stranded 3′ G-overhang (Trujillo et al, 2005). A conserved shelterin complex (composed of Taz1, Rap1, Poz1, Tpz1, Pot1, and Ccq1) binds to the telomeric DNA for chromosome end protection and telomere length regulation (de Lange, 2018). This highly ordered DNA-protein structure is essential for heterochromatin establishment at telomeres. Insertion of telomeric repeats at an internal chromosome position is able to induce the heterochromatin formation (Tadeo et al, 2013). Moreover, mutations of shelterin components Taz1, Rap1, Tpz1, or Ccq1 lead to de-repression of reporter genes inserted near telomeres, suggestive of an important role of the shelterin complex in heterochromatin establishment (Cooper et al, 1997; Kanoh & Ishikawa, 2001; Kanoh et al, 2005; Kallgren et al, 2014). It has been proposed that shelterin subunit Ccq1 mediates the interactions of shelterin with the SHREC (Snf2/histone deacetylase–containing

[1]State Key Laboratory of Oncogenes and Related Genes, Ninth People's Hospital, Shanghai Jiao Tong University School of Medicine, Shanghai, China   [2]Shanghai Institute of Precision Medicine, Shanghai, China   [3]Key Laboratory of Cell Differentiation and Apoptosis of Chinese Ministry of Education, Shanghai Jiao Tong University School of Medicine, Shanghai, China   [4]National Key Laboratory of Crop Genetic Improvement, Huazhong Agricultural University, Wuhan, China   [5]State Key Laboratory of Molecular Biology, Center for Excellence in Molecular Cell Science, Shanghai Institute of Biochemistry and Cell Biology, Chinese Academy of Sciences, Shanghai, China   [6]School of Life Science and Technology, ShanghaiTech University, Shanghai, China

Correspondence: leim@shsmu.edu.cn; zhenfwu@shsmu.edu.cn
*Shaohua Shi and Yuanze Zhou contributed equally to this work

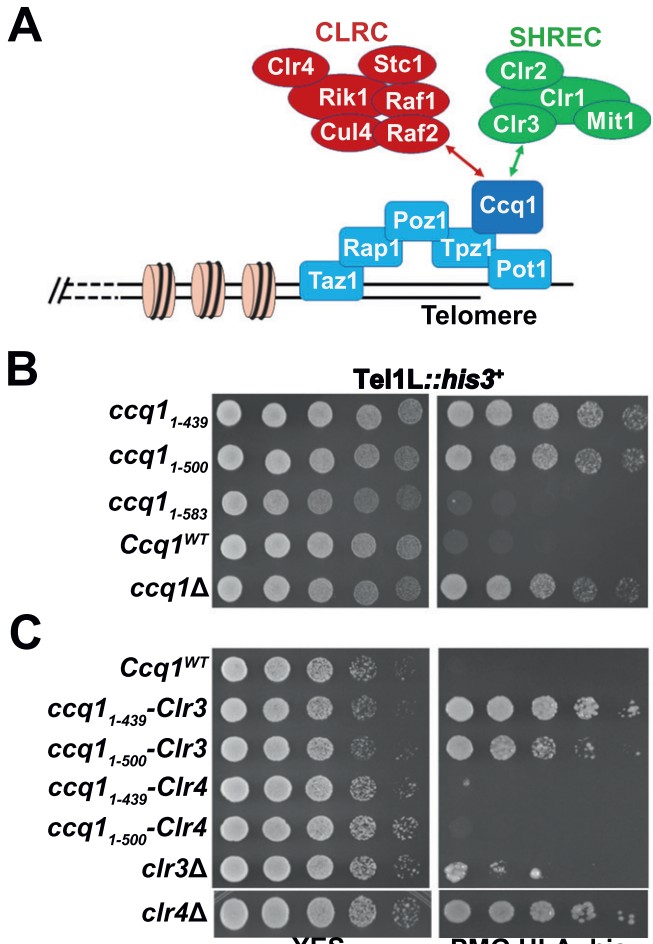

**Figure 1.  The CLRC complex plays a dominant in silencing telomeric heterochromatin.**
**(A)** A schematic diagram of the shelterin, CLRC, and SHREC complexes at telomeres. The Ccq1 subunit of shelterin facilitates recruitment of both the CLRC and the SHREC complexes to telomeres. **(B)** Effects of Ccq1 truncation mutants on the transcriptional silencing of $his3^+$ reporter gene inserted adjacent to the telomeric region. Equal amounts of 10-fold dilution series of cultures were spotted on YES or Pombe Medium Glutamate supplemented with uracil, leucine, and adenine (PMG ULA) (-histidine) plates. **(C)** Effects of the fusion of Ccq1$_{1-500}$ and Ccq1$_{1-439}$ mutants with Clr3 or Clr4 on the transcriptional silencing of $his3^+$. The strains were spotted on plates as in (B).

repressor complex, composed of Clr3, Mit1, Clr1, and Clr2) and the CLRC (Clr4 histone methyltransferase, composed of Clr4, Cul4, Rik1, Raf1, and Raf2) complexes for telomeric heterochromatin assembly (Fig 1A) (Sugiyama et al, 2007; Wang et al, 2016; van Emden et al, 2019). However, because shelterin also plays important roles in chromosome end protection and telomere length regulation, the mechanism of how the shelterin complex establishes heterochromatin at telomeres has remained elusive.

Here, we characterize the interaction between Ccq1 and CLRC component Raf2 and unveil the roles of the Ccq1–Raf2 interaction in shelterin-dependent recruitment of the CLRC complex and heterochromatin establishment at telomeres. We propose a positive feedback mechanism between shelterin and CLRC for the nucleation and spreading of subtelomeric heterochromatins.

# Results

## The CLRC complex plays a dominant role in transcriptional silencing at telomeres

To investigate the specific role of Ccq1 in telomeric heterochromatin establishment, we used two Ccq1 truncation mutants, Ccq1$_{1-500}$ and Ccq1$_{1-436}$. Consistent with previous data (Moser et al, 2015), both $ccq1_{1-500}$ and $ccq1_{1-436}$ cells failed to repress the expression of a $his^+$ reporter gene inserted adjacent to telomere IL comparable to $ccq1\Delta$ cells (Fig 1B). Because Ccq1$_{1-436}$ is sufficient for the interaction with Tpz1 (Jun et al, 2013; Harland et al, 2014; Armstrong et al, 2018), the failure of Ccq1$_{1-500}$ and Ccq1$_{1-436}$ truncation mutants in heterochromatin formation at telomeres might be due to defect in recruitment of the SHREC and/or CLRC complexes to telomeres. To address this issue, we fused the deacetylase subunit Clr3 of SHREC and the methyltransferase subunit Clr4 of CLRC to Ccq1$_{1-500}$ and Ccq1$_{1-436}$, respectively, and investigated the functions of the SHREC and CLRC complexes in telomeric heterochromatin establishment. Our results clearly showed that fusion of Clr4, but not Clr3, with either Ccq1$_{1-500}$ or Ccq1$_{1-436}$ could rescue the repression of $his3^+$ expression (Fig 1C), indicative of an important role of the CLRC complex in transcriptional silencing at telomeres. Therefore, unlike the essential roles of both SHREC and CLRC in heterochromatin assembly at centromeres and silent mating-type loci (Sugiyama et al, 2007; Zhang et al, 2008), heterochromatin nucleation at telomeres predominantly involves the CLRC complex. These results are also in accordance with the observation that $clr4\Delta$ cells exhibited much more obvious defects in the repression of $his3^+$ expression than the $clr3\Delta$ cells (Fig 1C).

## Identification of Ccq1 residues critical for the Ccq1–Raf2 interaction

The observation that $ccq1_{1-583}$ mutant cells maintained the transcriptional repression of $his3^+$ expression suggested that Ccq1 residues 500–583 are important for the recruitment of the CLRC complex to establish telomeric heterochromatin, which is consistent with the finding that Ccq1 mediates a direct interaction with CLRC subunit Raf2 (Fig 1B) (Moser et al, 2015; Wang et al, 2016). To map the interaction between Ccq1 and Raf2, we further truncated the Ccq1 protein and found that a smaller fragment Ccq1$_{496-583}$ is sufficient for the interaction with Raf2 (Fig 2A–C). This is in accordance with the data that Ccq1 residues 500–583 are responsible for telomeric heterochromatin establishment and gene silencing (Fig 1B). Hereafter, we will refer to Ccq1$_{496-583}$ as the Raf2-binding motif of Ccq1 (Ccq1$_{RBM}$) (Fig 2A).

Multiple sequence alignment analysis of Ccq1 proteins from various species revealed that there is a cluster of conserved hydrophobic residues in Ccq1$_{RBM}$ (Leu511, Val516, Tyr518, and Leu519) (Fig 2B). We found that arginine substitution of either of these residues could efficiently disrupt the interaction of Raf2 with both Ccq1$_{RBM}$ and full-length Ccq1 in yeast two-hybrid assays (Fig 2D and E). Notably, none of these mutations affected the interactions of Ccq1 with Tpz1 and Clr3 (Fig S1). Furthermore, co-immunoprecipitation (co-IP) experiments revealed that Ccq1$^{L511R}$, Ccq1$^{V516R}$, and Ccq1$^{Y518R}$ mutations

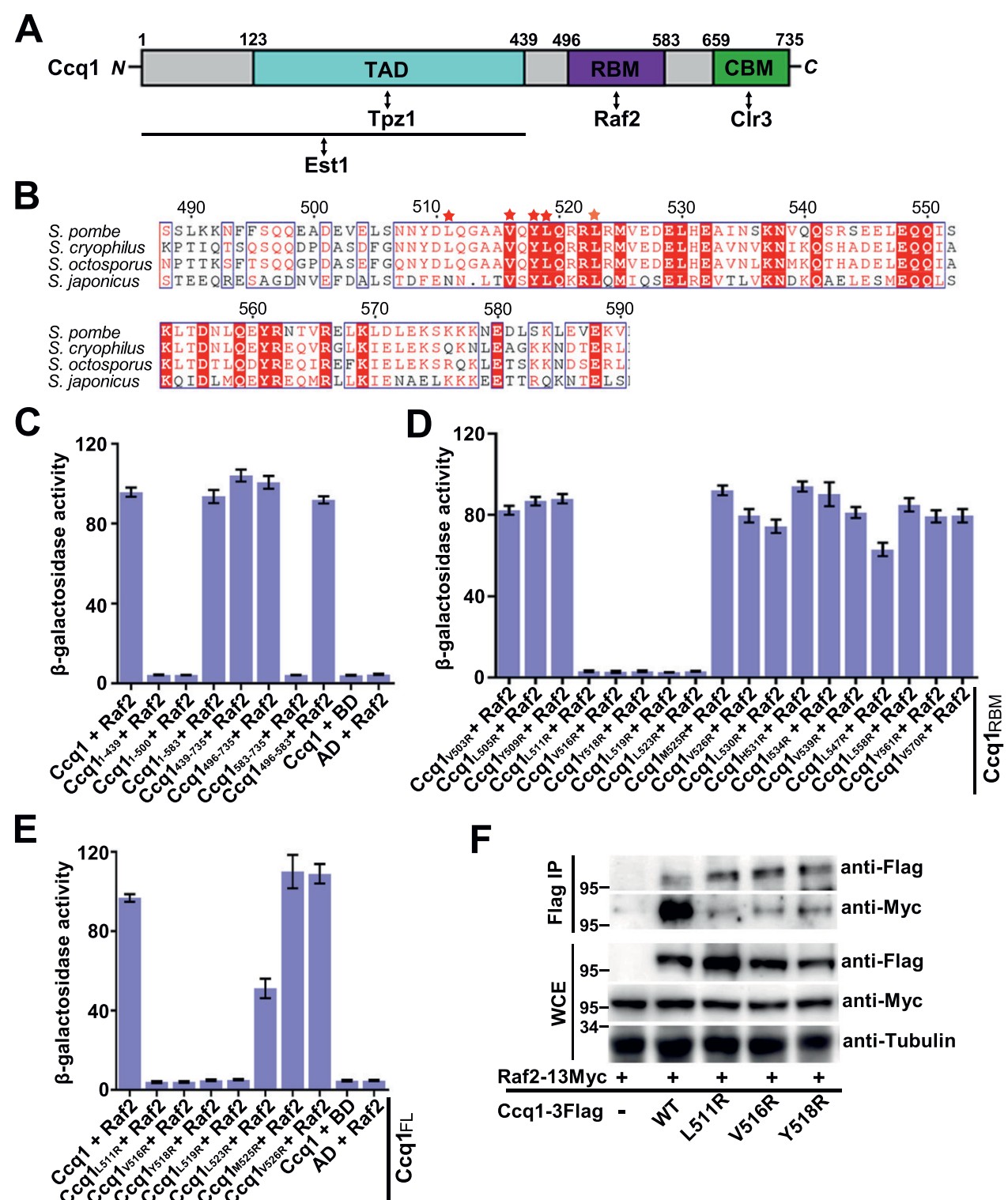

**Figure 2. Biochemical characterization of the Ccq1–Raf2 interaction.**
**(A)** Domain organization of Ccq1. The domains and motifs within Ccq1 that mediate interactions with Est1 (Moser et al, 2011), Tpz1 associating domain (TAD) (Jun et al, 2013; Harland et al, 2014), Raf2-binding motif (RBM) and Clr3-binding motif (CBM) (Armstrong et al, 2018) are designated. **(B)** Multiple sequence alignment of *Schizosaccharomyces pombe* Ccq1$_{RBM}$ and its homologues. Conserved residues of Ccq1$_{RBM}$ are boxed and highlighted in red. Red stars denote residues important for the Ccq1–Raf2 interaction. **(C)** Identification of the domain of Ccq1 that mediates interaction with Raf2 by yeast two-hybrid (Y2H) analysis. **(D)** Y2H assay to screen mutations of Ccq1$_{RBM}$ that disrupt the Ccq1$_{RBM}$–Raf2 interaction. **(E)** Effects of Ccq1 mutations on the interaction of full-length Ccq1 (Ccq1$_{FL}$) with Raf2 were examined in Y2H assays. **(C, D, E)** In (C, D, E), Raf2 was fused to Gal4 DNA-binding domain, and WT and mutant Ccq1 were individually fused to Gal4 activation domain. Error bars in the graph represent mean ± SEM. **(F)** Co-IP analysis of the interaction of Myc-tagged Raf2 with Flag-tagged wild-type or mutant Ccq1. The levels of each protein in input and IP samples were analyzed by immunoblotting with the indicated antibodies.
Source data are available for this figure.

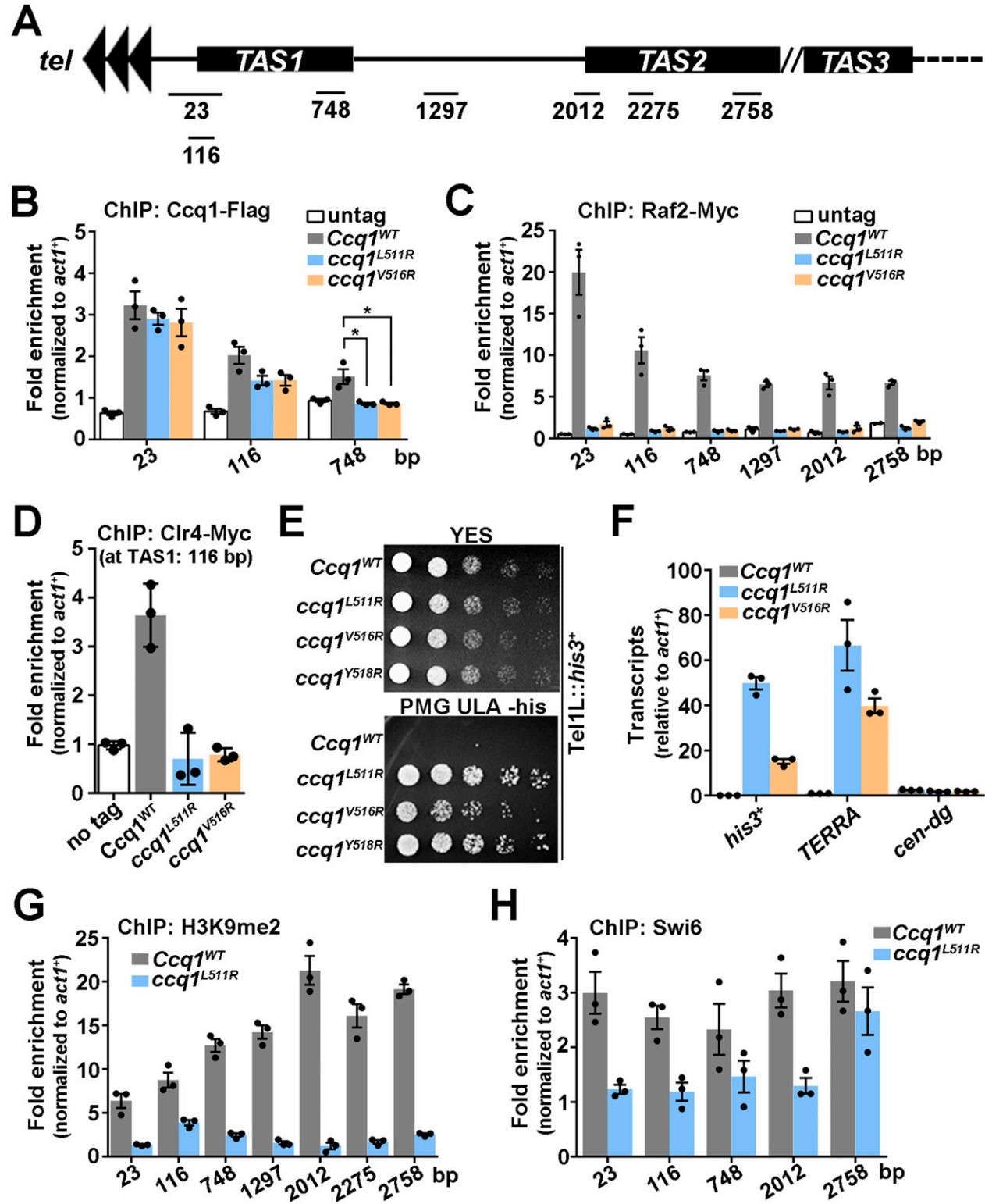

**Figure 3. The Ccq1-Raf2 interaction mediates heterochromatin formation and CLRC enrichment at telomeres.**
**(A)** A schematic figure of telomeric and subtelomeric regions in fission yeast *Schizosaccharomyces pombe*. Positions on x axis represent distances from telomeric repeats (van Emden et al, 2019). **(B, C, D)** ChIP-qPCR analysis of Ccq1 (B), Raf2 (C), and Clr4 (D) in WT, *ccq1^L511R^*, and *ccq1^V516R^* strains. Recruitment to the internal *act1^+^* locus serves as a control for ChIP specificity. **(E)** Effects of Raf2-binding deficient mutations of Ccq1 (Ccq1^L511R^ and Ccq1^V516R^) on the transcriptional silencing of *his3^+^* reporter gene inserted adjacent to the telomeric region. **(F)** RT-qPCR analysis of the transcription of *TERRA* in WT and *ccq1^L511R^* and *ccq1^V516R^* mutant strains, normalized to that of *act1^+^*. The transcription at the *cen-dg* region was used as a control. **(G, H)** ChIP-qPCR analysis of H3K9me2 (G) and Swi6 (H) in WT and *ccq1^L511R^* cells. Recruitment to the internal *act1^+^* locus serves as a control for ChIP specificity. **(B, C, D, F, G, H)** Data information: In (B, C, D, F, G, H), data are represented as means from three independent experiments. **(B)** In (B), the error bars represent mean ± SEM. *0.05 > *P* > 0.01 (*t* test). Source data are available for this figure.

maintained the expression and stability of Ccq1 protein (Fig S2A), but greatly weakened the Ccq1–Raf2 interaction in yeast cells (Fig 2F), suggesting that the correctly folded mutant Ccq1 proteins only specifically disrupt the interaction with Raf2. Taken together, our data demonstrated that hydrophobic residues L511, V516, Y518, and L519 in Ccq1$_{RBM}$ play a critical role in mediating the specific interaction between Ccq1 and Raf2.

### The Ccq1–Raf2 interaction contributes to the association of CLRC with telomeres and subtelomeres

To investigate the functional significance of the Ccq1–Raf2 interaction, we asked whether Raf2-binding deficient mutants Ccq1$^{L511R}$ and Ccq1$^{V516R}$ affected the association of the CLRC complex with telomeres in vivo. Chromatin immunoprecipitation (ChIP) analysis showed that the enrichment of Ccq1 at either telomeric repeats or telomere-proximal telomere-associated sequences (TAS) was modestly but reproducibly decreased in $ccq1^{L511R}$ and $ccq1^{V516R}$ cells (Fig 3A and B). Notably, our ChIP data revealed that Raf2 was localized at both telomeric repeats and subtelomeric TAS (Fig 3A and C). Moreover, Raf2 signals at all tested regions were completely eliminated after the disruption of the Ccq1–Raf2 interaction in $ccq1^{L511R}$ and $ccq1^{V516R}$ cells (Fig 3A and C). It was reported that the shelterin complex is enriched at telomeres and subtelomeres (Inoue et al, 2019; van Emden et al, 2019). These results suggest that the Ccq1–Raf2 interaction contributes to the association of Raf2 with telomeric and subtelomeric regions. Furthermore, our ChIP data for the telomere association of the methyltransferase subunit Clr4 of CLRC showed a great reduction of Clr4 at telomeres in both $ccq1^{L511R}$ and $ccq1^{V516R}$ cells (Fig 3D), suggestive of shelterin-mediated CLRC recruitment to chromosomal ends by the Ccq1–Raf2 interaction.

To exclude the impact of Raf2-binding deficient Ccq1 mutants on telomerase recruitment and activation, we first analyzed the amount of telomerase RNA TER1 that was coimmunoprecipitated with WT or mutant Ccq1. RNA-immunoprecipitation (RIP) data showed that both Ccq1$^{L511R}$ and Ccq1$^{V516R}$ mutant proteins coimmunoprecipitated with TER1 at comparable levels to that of WT cells (Fig S2B). In addition, telomere Southern blot analysis revealed that the $ccq1^{L511R}$ and $ccq1^{V516R}$ mutant cells maintained normal telomere length (Fig S2C), consistent with a previous study that the Ccq1 C-terminal residues 501–735 are dispensable for telomere-length maintenance (Moser et al, 2015). Moreover, unlike the situation that shortened telomeres lead to highly elongated cells indicative of checkpoint activation in $ccq1\Delta$ and Tpz1-Ccq1 interaction deficient cells (Moser et al, 2011, 2015; Harland et al, 2014), cell elongation was not observed in either $ccq1^{L511R}$ or $ccq1^{V516R}$ mutant strains (Fig S2D). These results indicated that disruption of the Ccq1–Raf2 interaction has no effect on Ccq1-dependent telomerase recruitment and telomere maintenance, in accordance with the fact that Raf2-binding deficient Ccq1 mutations are outside of the Tpz1- and Est1-binding domains of Ccq1 (Fig 2A). Taken together, by taking advantage of the separation-of-function Ccq1 mutants that specifically disrupt the Ccq1–Raf2 interaction we demonstrated that the Ccq1–Raf2 interaction contributes to the shelterin-mediated recruitment of the CLRC complex to telomeres and subtelomeres.

### The Ccq1–Raf2 interaction contributes to telomeric heterochromatin formation

To investigate the in vivo function of the Ccq1–Raf2–dependent CLRC recruitment to telomeres, we analyzed the effects of Ccq1$^{L511R}$ and Ccq1$^{V516R}$ mutations on transcriptional silencing at telomeres. Our results showed that both of the $ccq1^{L511R}$ and $ccq1^{V516R}$ mutant strains failed to repress the expression of the $his3^+$ reporter gene (Fig 3E), indicating that disruption of the Ccq1–Raf2 interaction led to a defect in transcriptional silencing at telomeres. Next, we examined the expression of TERRA (telomeric repeat-containing non-coding RNA) by reverse transcription-quantitative PCR (RT-qPCR), which clearly showed that transcripts of TERRA were substantially increased in $ccq1^{L511R}$ and $ccq1^{V516R}$ cells compared to WT cells (Fig 3F). As a control, transcription of the centromeric region (cen-dg) was unaffected in $ccq1^{L511R}$ and $ccq1^{V516R}$ cells (Fig 3F). Collectively, we conclude that the Ccq1–Raf2–mediated recruitment of the CLRC complex contributes to transcriptional silencing of both reporter gene and endogenous non-coding RNA at subtelomeres.

Heterochromatic regions are associated with high levels of methylation of histone H3 at Lys9 (H3K9me) and heterochromatin protein Swi6 (Grewal & Jia, 2007). Consistently, our ChIP data showed that the levels of both H3K9me2 and Swi6 at subtelomeric regions were markedly decreased in $ccq1^{L511R}$ cells (Fig 3A, G, and H), further supporting the notion that telomeric heterochromatin structures are impaired by the disruption of the Ccq1–Raf2 interaction. Taken together, we conclude that the Ccq1–Raf2 interaction plays a crucial role in shelterin-dependent recruitment of the CLRC complex and heterochromatin establishment at subtelomeres.

### The Ccq1–Raf2 interaction promotes nucleosome stability and shelterin association with telomeres

To exclude the possibility that the reduction of H3K9me2 levels at subtelomeric regions is due to the low nucleosome occupancy in Ccq1–Raf2 interaction deficient cells, we determined the total H3 levels in WT and $ccq1^{L511R}$ cells by ChIP assay. Our results revealed a modest decrease in histone H3 levels at telomere-proximal TAS regions in $ccq1^{L511R}$ cells (Figs 3A and 4A), in accordance with previous data that CLRC association promotes nucleosome stability at subtelomeres (van Emden et al, 2019). It should be noted that the relative decrease in the H3K9me2 signal in WT versus $ccq1^{L511R}$ cells was far more pronounced than that of H3 (Fig 4B), suggesting that the loss of CLRC-mediated methyltransferase activity, rather than low histone occupancy, likely is the major reason for the reduction in H3K9me levels at subtelomeres after the disruption of the Ccq1–Raf2 interaction. Collectively, we propose that the association of the CLRC complex at subtelomeres promotes both H3K9 methylation and nucleosome stability, and the nucleosome stability in turn further facilitates the methylation of H3K9 by CLRC's methyltransferase activity. The H3K9 methylation is required for sequential recruitment of Swi6 to nucleate heterochromatin.

It has been reported that the CLRC complex also promotes the shelterin-chromatin association at subtelomeres independent of heterochromatin formation (van Emden et al, 2019). Consistently, we observed that the telomere association of Ccq1 was reduced after the disruption of the Ccq1–Raf2 interaction (Fig 3B). Next, we

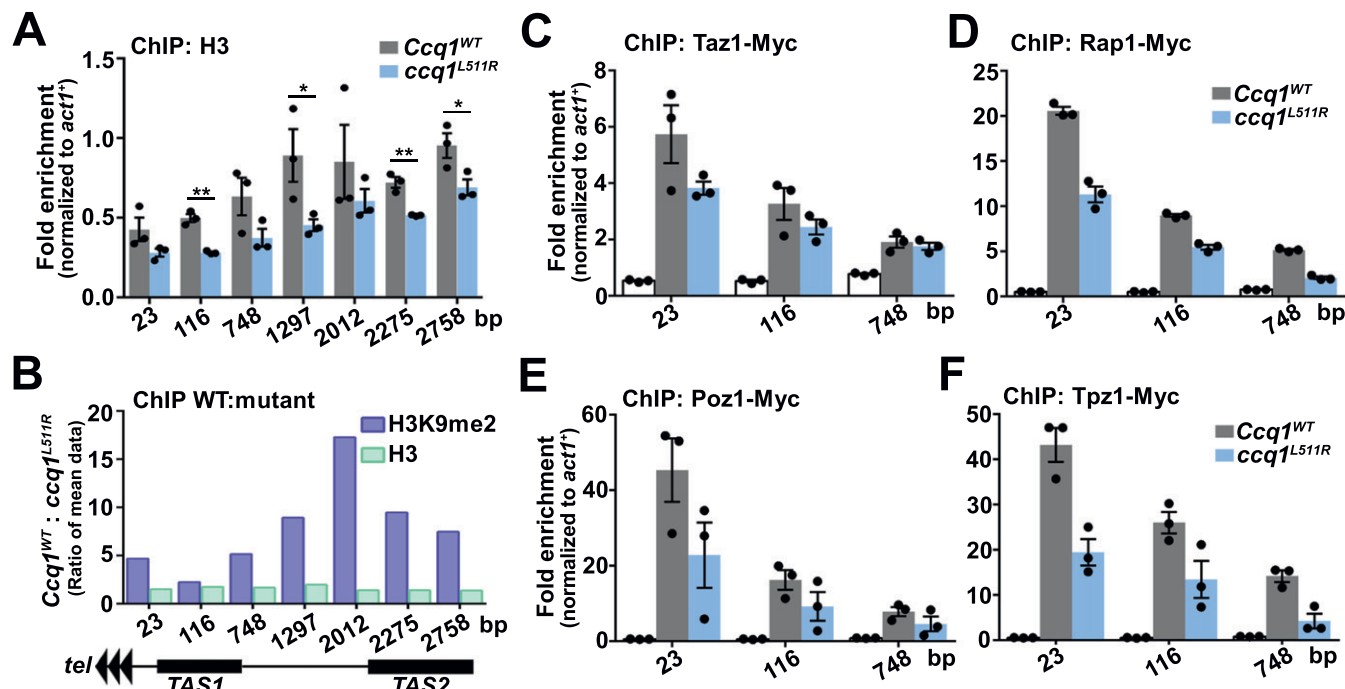

**Figure 4.  Ccq1-CLRC-mediated heterochromatin promotes nucleosome stability and shelterin enrichment at subtelomeres.**
**(A)** ChIP-qPCR analysis of histone H3 in WT and $ccq1^{L511R}$ cells. Recruitment to the internal $act1^+$ locus serves as a control for ChIP specificity. **(B)** Fold change of WT over $ccq1^{L511R}$ for H3K9me2 and H3 levels. **(C, D, E, F)** Effects of the $Ccq1^{L511R}$ mutation on telomere association for Taz1 (C), Rap1 (D), Poz1 (E) and Tpz1 (F) were measured by ChIP-qPCR assays. Recruitment to the internal $act1^+$ locus serves as a control for ChIP specificity. **(A, C, D, F)** Data information: In (A, C, D, E, F), data are represented as means from three independent experiments. **(A)** In (A), the error bars represent mean ± SEM. *0.05 > $P$ > 0.01; **$P$ < 0.01 ($t$ test).
Source data are available for this figure.

examined whether the Ccq1$^{L511R}$ mutant affected the subtelomeric association of other shelterin protein factors. Our ChIP data clearly showed that the associations of Taz1, Rap1, Poz1, and Tpz1 were all decreased at telomeric repeats and the subtelomeric TAS in $ccq1^{L511R}$ cells (Fig 4C–F), supporting the notion that CLRC promotes shelterin-chromatin association (van Emden et al, 2019). Taken together, we propose a feedback loop mechanism between the shelterin and the CLRC complexes, in which shelterin recruits CLRC via the Ccq1–Raf2 interaction, whereas CLRC in turn promotes the shelterin association with subtelomeric chromatins that facilitates the recruitment of more CLRC complexes.

## Discussion

Heterochromatin is implicated in multiple chromatin-associated processes, such as gene regulation, chromosome segregation, and suppression of homologous recombination to ensure genome integrity (Allshire et al, 1995; Allshire & Madhani, 2018). Because of the highly conserved heterochromatin machinery between fission yeast and higher eukaryotes, *S. pombe* has been serving as a model organism for mechanistic studies of the heterochromatin assembly at different chromosomal regions (Grewal & Jia, 2007; Allshire & Madhani, 2018). The heterochromatin assembly pathways at centromeres and silent mating-type loci have been well-characterized in *S. pombe* during the past two decades (Grewal & Jia, 2007). Multiple lines of evidence have revealed that the shelterin complex is

required for heterochromatin maintenance at telomeres. However, the mechanism involved in this process remains an open question because of the multiple functions of shelterin in telomere biology (de Lange, 2018). In this study, we demonstrate that the interaction between shelterin protein Ccq1 and Raf2 mediates CLRC recruitment and heterochromatin assembly at telomeres.

Heterochromatin nucleation by RNAi- and DNA-mediated pathways at centromeres and silent mating-type loci requires the recruitment of both HMTs and HDACs in fission yeast (Sugiyama et al, 2007; Zhang et al, 2008). Here, however, we show that heterochromatin nucleation at telomeres predominantly involves the CLRC but not the SHREC complex. We provide an integrated picture for subtelomeric heterochromatin nucleation by the shelterin complex in fission yeast (Fig 5). In this model, shelterin recruits CLRC to telomeric repeats and the subtelomeric regions via the Ccq1–Raf2 interaction. The CLRC enrichment promotes histone H3K9 methylation, nucleosome stability, as well as shelterin–chromatin association at subtelomeres. The nucleosome stability further facilitates methylation of H3K9 by CLRC and the association of CLRC in turn promotes shelterin association with chromatin that recruits more CLRC to telomeres. This positive feedback loop between the shelterin and the CLRC complexes plays a critical role in the nucleation and spreading of heterochromatin at subtelomeres (Fig 5).

The Ccq1-SHREC and Ccq1-CLRC interactions have been proposed to function in the repression of telomere elongation after Ccq1–Est1–mediated telomerase recruitment and activation in fission yeast (Tomita & Cooper, 2008; Armstrong et al, 2018). Thus, an outstanding

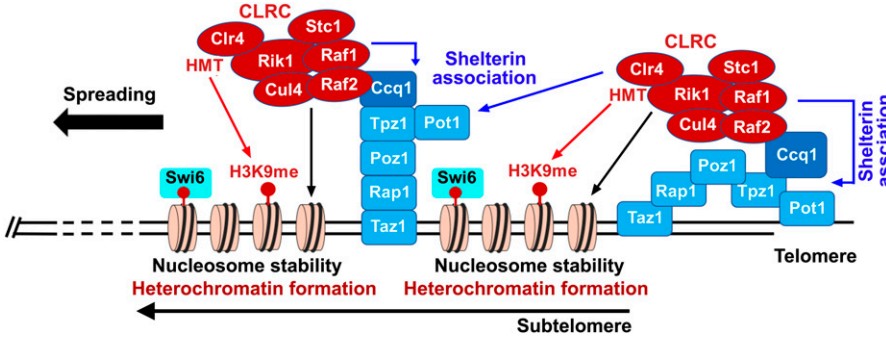

**Figure 5. A schematic model for the nucleation and spreading of heterochromatin at subtelomeres by the shelterin and CLRC complexes.**
The Ccq1-Raf2 interaction contributes to CLRC association with telomeric repeats and subtelomeric regions that catalyzes the histone H3K9 methylation, and the methylated histone H3K9 sequentially recruit Swi6 to initiate heterochromatin formation. Moreover, The CLRC association promotes nucleosome stability at subtelomeres, which further facilitates the methylation of histone H3K9 by CLRC. Finally, shelterin-mediated CLRC recruitment in turn facilitates the shelterin association with subtelomeric chromatin, and this positive feedback loop between the shelterin and the CLRC complexes plays a critical role in the nucleation and spreading of heterochromatin at subtelomeres.

question is how Ccq1 coordinates its multiple activities at telomeres. We propose that these processes are likely coupled together through conformational changes induced by Ccq1 interactions with different complexes in a highly orchestrated manner. Notably, *cenH*-like repeats have been identified in the putative telomere-linked helicase (*tlh*$^+$) genes, ~15 kb away from telomere ends (Hansen et al, 2006), indicating that, similar to what happens at the silent mating-type loci, both RNAi- and DNA-based heterochromatin assembly mechanisms might be adopted at subtelomeres as well. Future studies are required to fully understand how shelterin and RNAi-based pathway coordinate to establish heterochromatin at eukaryotic chromosomal ends.

# Materials and Methods

### Strains, gene tagging, and mutagenesis

The growth media and basic genetic techniques were performed as previously described (Forsburg & Rhind, 2006; Lorenz, 2015). The yeast strain TN9125, carrying an integrated *his3*$^+$ marker adjacent to telomeric repeats of the chromosome IL, was a gift from Dr Toru M Nakamura (Nimmo et al, 1998). Genes tagged with the 13×Myc or 3×FLAG epitope was introduced as described (Gadaleta et al, 2013). Mutations in the *ccq1*$^+$ gene with kanMx6 (*kan*$^r$) were created by PCR, and each mutated DNA fragment was integrated at the endogenous gene's locus. All strains used in this study are listed in Table S1.

### Yeast growth on plates

Single colonies were inoculated into 5 ml of yeast extract with supplement (YES) and cultured to saturation. The cultures were then diluted to OD$_{600}$ = 1, and equal amounts (5 μl) of 10-fold serial dilutions of the cultures were spotted on YES or Pombe Medium Glutamate supplemented with uracil, leucine, and adenine (PMG ULA) (–histidine) plates. After incubation at 30°C for 2–3 d, plates were photographed.

### Yeast two-hybrid assay

The yeast two-hybrid assay was performed as described previously (Xue et al, 2017). Briefly, the L40 strain was transformed with pBTM116 and pACT2 (Clontech) fusion plasmids, and colonies harboring

both plasmids were selected on Yeast complete–Leu–Trp plates. The β-galactosidase activities were measured by a liquid assay.

### RT-qPCR analysis

Total RNA was isolated using RNeasy mini kit (QIAGEN). 1 μg RNA was used as template for the reverse transcription of 20 μl cDNA using PrimeScript RT reagent Kit with gDNA Eraser (Perfect Real Time) (TAKARA). 2 μl of the RT reaction were used to analyze gene expression level by quantitative real-time PCR and normalized to that of *act1*$^+$. The real-time PCR was performed in the LightCycler 480 (Roche), and the TB Green *Premix Ex Taq* II (Tli RNaseH Plus) (TAKARA) reagent was used. The qPCR conditions were 30 s at 95°C, 40 cycles of 5 s at 95°C for denaturation, 30 s at 60°C for annealing and extension. The primers were used as described previously (Table S2) (Braun et al, 2011; Bah et al, 2012; van Emden et al, 2019).

### Co-immunoprecipitation (co-IP) and Western blot analysis

Co-IP experiments were performed as described previously (Harland et al, 2014). Whole-cell extracts were prepared in lysis buffer (50 mM Hepes, pH 7.5, 150 mM NaCl, 1 mM EDTA, 1% Triton-X100, and complete protease inhibitor cocktail [Roche]). Cell lysates were centrifuged and supernatants were precleared and immunoprecipitated with anti-FLAG M2 Affinity Gel (Sigma-Aldrich) at 4°C with rocking for 4 h. Precipitates were then washed with lysis buffer and subjected to SDS–PAGE separation. After SDS–PAGE, proteins were blotted onto polyvinylidene fluoride (PVDF) membranes (Millipore). The blots were incubated in blocking buffer (5% fat-free milk in PBS buffer supplemented with 0.05% TWEEN-20) at RT for 1 h and incubated with primary antibodies in blocking buffer at 4°C for overnight. Blots were then washed and incubated in the HRP-labeled secondary antibodies at RT for 1 h. After wash, blots were developed with ECL Prime Western Blotting System (RPN2232; GE Healthcare).

### Telomere southern blot

Telomere blot was performed as described previously (Jun et al, 2013; Harland et al, 2014; Moser et al, 2015). Briefly, *ccq1* mutant transformants were confirmed by PCR and sequencing. The cells were harvested from 5 ml liquid culture inoculated from YES plates. Genomic DNA was purified by using phenol chloroform method,

digested with *Eco*R I, and fractionated by electrophoresis on 1.0% agarose gel. The DNA fragments were transferred to a Hybond-N⁺ Nylon membrane (GE Healthcare), UV cross-linked and incubated with Church buffer for 30 min at 50°C. Biotinylated telomeric-specific probe was incubated with the DNA at 50°C overnight, and biotin probe–bound DNA fragments corresponding to telomeric DNA were detected using Chemiluminescent Nucleic Acid Detection Module (Thermo Fisher Scientific).

### Chromatin immunoprecipitation (ChIP) assay

The ChIP assay was performed as described previously (Moser et al, 2015; Ge et al, 2020). Yeast cells in exponential growth phase were diluted to the same cell density, crosslinked for 20 min with 1% formaldehyde, and quenched with 125 mM glycine for 10 min. Cells were pelleted and washed twice with 20 ml ice-cold PBS buffer and once with pre-chilled lysis buffer (50 mM Hepes, pH 7.5, 150 mM NaCl, 1 mM EDTA, 1% Triton X-100, and 0.1% sodium deoxycholate). The cell pellets were re-suspended in 500 $\mu$l lysis buffer containing 5 $\mu$l cocktail and 5 $\mu$l PMSF. Cells were lysed by using acid-washed glass beads, and then 250 $\mu$l of cell extracts were sonicated (pulse on 30 s, pulse off 30 s, 20 cycles) in a pre-chilled Bioruptor (Diagenode) to obtain chromatin fragments of about 300–500 bp in size. The soluble chromatin was obtained by centrifugation at full speed for 10 min. A 10 $\mu$l of the ChIP extract was taken for immunoprecipitation (IP) input, and 1.25 $\mu$l (1:200) of indicated antibodies (anti-Myc or anti-FLAG) were added to the remaining chromatin extract. Protein A Sepharose 4 Fast Flow beads (GE Healthcare) were washed three times with lysis buffer, and added to the ChIP extracts. After incubation at 4°C for 4–6 h, beads were washed once with lysis buffer, buffer I (50 mM Hepes, pH 7.5, 500 mM NaCl, 1 mM EDTA, 1% Triton-X100, and 0.1% sodium deoxycholate), buffer II (10 mM Tris–HCl, pH 8.0, 0.25 M LiCl, 1 mM EDTA, NP-40, and 0.5% sodium deoxycholate) and TE (10 mM Tris–HCl, pH 8.0, and 1 mM EDTA) each for 5 min. Bead-bound DNAs were eluted in 150 $\mu$l TE/1% SDS at 70°C for 30 min. IPs and inputs were incubated at 65°C overnight for reverse-crosslinking, and DNAs were purified with QIAquick PCR Purification Kit (QIAGEN). The real-time qPCR analysis was performed in the LightCycler 480 (Roche), and the TB Green Premix Ex Taq II (Tli RNaseH Plus) (TAKARA) reagent was used. The qPCR conditions were 30 s at 95°C, 40 cycles of 5 s at 95°C for denaturation, 30 s at 55°C for annealing and 30 s at 72°C for extension. Telomere enrichment was calculated as fold change of telomere product normalized to *act1⁺* locus product with the following formula $2^{[(Ct\ Act\ IP\ -\ Ct\ Act\ Input)\ -\ (Ct\ Tel\ IP\ -\ Ct\ Tel\ Input)]}$. The primers targeting telomeric and subtelomeric regions were used as described (Table S2) (Harland et al, 2014; van Emden et al, 2019).

### Co-IP of TER1 RNA and Ccq1

The co-IP of TER1 and Ccq1 was performed as described (Harland et al, 2014; Collopy et al, 2018). Briefly, cells were pelleted and washed twice with 20 ml ice-cold PBS buffer and once with pre-chilled lysis buffer (50 mM Hepes, pH 7.5, 150 mM NaCl, 1 mM EDTA, 1% Triton X-100, and 0.1% sodium deoxycholate). The cell pellets were re-suspended in 500 $\mu$l lysis buffer containing 5 $\mu$l cocktail, 5 $\mu$l PMSF, and 40 U/ml RNAase inhibitor. Cells were lysed by using acid-washed

glass beads, and the supernatant was obtained by centrifugation at full speed for 10 min. Co-IP of Flag-tagged Ccq1 and TER1 was performed with anti-Flag M2 antibody (Sigma-Aldrich) and protein A Sepharose beads (GE healthcare). The RNA on the beads was purified using RNeasy mini kit (QIAGEN), which was used as template for the reverse transcription using PrimeScript RT reagent Kit with gDNA Eraser (Perfect Real Time) (TAKARA). The amount of TER1 was quantified using real-time qPCR analysis in the LightCycler 480 (Roche), and the TB Green Premix Ex Taq II (Tli RNaseH Plus) (TAKARA) reagent was used. The qPCR conditions were 30 s at 95°C, 40 cycles of 5 s at 95°C for denaturation, 30 s at 55°C for annealing, and 30 s at 72°C for extension. Primers for TER1 were used as described (Table S2) (Collopy et al, 2018). % Precipitated TER1 RNA values were calculated based on ΔCt between Input and IP samples.

## Data Availability

No data were deposited in a public database.

## Supplementary Information

## Acknowledgements

This work was supported by grants from the Ministry of Science and Technology of China (2018YFA0107004 to M Lei), the National Natural Science Foundation of China (31930063 to M Lei).

### Author Contributions

S Shi: data curation, investigation, and methodology.
Y Zhou: data curation and methodology.
Y Lu: data curation and investigation.
H Sun: formal analysis.
J Xue: formal analysis.
Z Wu: conceptualization, data curation, investigation, and writing—original draft.
M Lei: conceptualization, supervision, funding acquisition, project administration, and writing—review and editing.

### Conflict of Interest Statement

The authors declare that they have no conflict of interest.

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
