## [Reviewer comments · Life Science Alliance]

Life Science Alliance

Ccq1-Raf2 interaction mediates CLRC recruitment to establish heterochromatin at telomeres

Ming Lei, Shaohua Shi, Yuanze Zhou, Yanjia Lu, Hong Sun, Jing Xue, and Zhenfang Wu
DOI: <https://doi.org/10.26508/lsa.202101106>

Corresponding author(s): Ming Lei, Shanghai Ninth People's Hospital, Shanghai Jiao Tong University School of Medicine

Review Timeline:

Submission Date:	2021-04-23
Editorial Decision:	2021-06-21
Revision Received:	2021-08-19
Editorial Decision:	2021-08-24
Revision Received:	2021-08-25
Accepted:	2021-08-25

Transaction Report:

June 21, 2021

Re: Life Science Alliance manuscript #LSA-2021-01106-T

Prof. Ming Lei
Shanghai Ninth People's Hospital, Shanghai Jiao Tong University School of Medicine
Shanghai Institute of Precision Medicine
115 Jinzun Road
Shanghai 200125
China

Dear Dr. Lei,

Thank you for submitting your manuscript entitled "Ccq1-Raf2 interaction mediates CLRC recruitment to establish heterochromatin at telomeres in fission yeast" to Life Science Alliance. The manuscript was assessed by an expert reviewer, whose comments are appended to this letter. We invite you to submit a revised manuscript addressing the Reviewer comments.

When submitting the revision, please include a letter addressing the reviewer comments point by point.

Thank you for this interesting contribution to Life Science Alliance. We are looking forward to receiving your revised manuscript.

Sincerely,

Eric Sawey, PhD
Executive Editor
Life Science Alliance

- A letter addressing the reviewer comments point by point.
- An editable version of the final text (.DOC or .DOCX) is needed for copyediting (no PDFs).
- High-resolution figure, supplementary figure and video files uploaded as individual files: See our detailed guidelines for preparing your production-ready images, <https://www.life-science-alliance.org/authors>
- Summary blurb (enter in submission system): A short text summarizing in a single sentence the study (max. 200 characters including spaces). This text is used in conjunction with the titles of papers, hence should be informative and complementary to the title and running title. It should describe the context and significance of the findings for a general readership; it should be written in the present tense and refer to the work in the third person. Author names should not be mentioned.

B. MANUSCRIPT ORGANIZATION AND FORMATTING:

Reviewer #1 (Comments to the Authors (Required)):

This manuscript details the interaction between Ccq1 and Raf2 that governs the recruitment of the CLRC complex to telomeres. Ccq1 is a component of telomere-bound Shelterin and Raf2 is a component of the Clr4 methyltransferase complex CLRC. The authors identify an interaction domain by truncations of Ccq1 and then identify critical residues within Ccq1 by comparing conserved sequences in the domain. Point mutants of the domain disrupt the Ccq1-Raf2 interaction. They also reduce subtelomeric silencing, H3K9me and H3 density, as well as Shelterin recruitment. The work is basically acceptable for publication after addressing the following points.

1) The RIP experiment in figure EV2B shows that TER1 binding levels are unchanged by the Ccq1 mutations. This shows that telomerase recruitment is unchanged. It does not show that telomerase is equally functional in the Ccq1 mutant. Since Ccq1 modulates telomerase activity (Armstrong,

Tomita, et al. 2018), this is a reasonable concern. A Southern should be performed to analyze telomeric ends, as was done in the first figure of the Armstrong paper, to determine whether the authors have identified a true separation-of-function mutant.

2) Figure legend 2A shows a graphic map of interaction motifs. Provide a reference to at least one paper for each motif that was identified before this manuscript.

3) Figure legend 2C and others describe yeast two hybrid analyses. Identify which protein was used as bait and which was used as prey.

3) Figure legend 3A and others show an X axis of numbers. The figures also show a graphic map of a telomere end. The numbers are likely the distances from the chromosome end but this needs to be clarified. It would also help to see graphically where the numbers correspond to on the graphic map of the telomere.

4) Figure legend 3A - Describe the statistical test that indicates these differences binding differences are significant.

5) Figure legend 3C needs to identify where Clr4 binding was measured.

6) Figure 3E RNA levels - The text says that cen-dg was included as a control site. Are all RNA levels measured relative to cen-dg or were they measured relative to another internal control?

7) Figure 4A - Describe the statistical test that indicates these differences binding differences are significant.

Point-by-point Responses to Reviewers' Comments

Reviewer #1 (Comments to the Authors (Required)):

This manuscript details the interaction between Ccq1 and Raf2 that governs the recruitment of the CLRC complex to telomeres. Ccq1 is a component of telomere-bound Shelterin and Raf2 is a component of the Clr4 methyltransferase complex CLRC. The authors identify an interaction domain by truncations of Ccq1 and then identify critical residues within Ccq1 by comparing conserved sequences in the domain. Point mutants of the domain disrupt the Ccq1-Raf2 interaction. They also reduce subtelomeric silencing, H3K9me and H3 density, as well as Shelterin recruitment. The work is basically acceptable for publication after addressing the following points.

Thanks!

1) The RIP experiment in figure EV2B shows that TER1 binding levels are unchanged by the Ccq1 mutations. This shows that telomerase recruitment is unchanged. It does not show that telomerase is equally functional in the Ccq1 mutant. Since Ccq1 modulates telomerase activity (Armstrong, Tomita, et al. 2018), this is a reasonable concern. A Southern should be performed to analyze telomeric ends, as was done in the first figure of the Armstrong paper, to determine whether the authors have identified a true separation-of-function mutant.

Thanks for pointing out this important issue. Following this reviewer's suggestion, we have performed telomere Southern blot to analyze telomere lengths of WT, *ccq1*^{L511R} and *ccq1*^{V516R} cells. Our result revealed that both Ccq1^{L511R} and Ccq1^{V516R} mutations maintained the WT telomere length (the revised Fig S2C). Moreover, microscopic analysis showed that Ccq1^{L511R} and Ccq1^{V516R} mutations did not result in cell elongation indicative of checkpoint activation (the revised Fig S2D). Thus, our results strongly support that we have identified true separation-of-function mutants that specifically disrupt the Ccq1-Raf2 interaction. We have modified the manuscript and Fig S2C and D accordingly.

“In addition, telomere Southern blot analysis revealed that the *ccq1*^{L511R} and *ccq1*^{V516R} mutant cells maintained normal telomere length (Fig S2C), consistent with a previous study reporting that the Ccq1 C-terminal residues 501-735 are dispensable for telomere-length maintenance (Moser *et al*, 2015). Moreover, unlike the situation that shortened telomeres lead to highly elongated cells indicative of checkpoint activation in *ccq1Δ* and Tpz1-Ccq1 interaction deficient cells (Harland *et al*, 2014; Moser *et al*, 2011; Moser *et al.*, 2015), cell elongation was not observed in either *ccq1*^{L511R} or *ccq1*^{V516R} mutant strains (Fig S2D).”

2) Figure legend 2A shows a graphic map of interaction motifs. Provide a reference to

at least one paper for each motif that was identified before this manuscript.

We have added the references in the revised Figure legend 2A.

3) Figure legend 2C and others describe yeast two hybrid analyses. Identify which protein was used as bait and which was used as prey.

Throughout this study, Ccq1 truncations or mutations were fused to the Gal4 activation domain (AD, prey), and other constructs were individually fused to Gal4 DNA binding domain (BD, bait). Following this reviewer's suggestion, we have replaced the "control" with "AD" or "BD" in the revised Fig 2C-2E and Fig S1. Furthermore, we have described this point in the revised figure legends of Fig 2C-2E and Fig S1.

4) Figure legend 3A and others show an X axis of numbers. The figures also show a graphic map of a telomere end. The numbers are likely the distances from the chromosome end but this needs to be clarified. It would also help to see graphically where the numbers correspond to on the graphic map of the telomere.

Thanks for this good point. we have added the graphic map of a telomere end labeled with the positions of primers (the distances from the chromosome end) in the revised Fig 3A.

5) Figure legend 3A - Describe the statistical test that indicates these differences binding differences are significant.

Following this reviewer's suggestion, we have added the statistical analysis using a t-test (*, $0.05 > p > 0.01$) in the revised Fig 3B.

6) Figure legend 3C needs to identify where Clr4 binding was measured.

The revised Fig 3D shows Clr4 binding at TAS1 (116 bp from telomeric repeats). We have added this point in the revised Fig 3D.

7) Figure 3E RNA levels - The text says that cen-dg was included as a control site. Are all RNA levels measured relative to cen-dg or were they measured relative to another internal control?

All RNA levels were normalized to *act1*⁺ in the RT-qPCR analysis (the "RT-qPCR analysis" of the "Materials and Methods" section). We have described this in the revised Fig 3F and figure legend.

8) Figure 4A - Describe the statistical test that indicates these differences binding differences are significant.

Following this reviewer's suggestion, we have added the statistical analysis using a t-test (*: $0.05 > p > 0.01$; **: $p < 0.01$) in the revised Fig 4A.

Reference

Harland JL, Chang YT, Moser BA, Nakamura TM (2014) Tpz1-Ccq1 and Tpz1-Poz1 interactions within fission yeast shelterin modulate Ccq1 Thr93 phosphorylation and telomerase recruitment. *PLoS Genet* 10: e1004708

Moser BA, Chang YT, Kosti J, Nakamura TM (2011) Tel1/ATR and Rad3/ATR kinases promote Ccq1-Est1 interaction to maintain telomeres in fission yeast. *Nat Struct Mol Biol* 18: 1408-1413

Moser BA, Raguimova ON, Nakamura TM (2015) Ccq1-Tpz1/TPP1 interaction facilitates telomerase and SHREC association with telomeres in fission yeast. *Mol Biol Cell* 26: 3857-3866

August 24, 2021

RE: Life Science Alliance Manuscript #LSA-2021-01106-TR

Prof. Ming Lei
Shanghai Ninth People's Hospital, Shanghai Jiao Tong University School of Medicine
Shanghai Institute of Precision Medicine
115 Jinzun Road
Shanghai, Shanghai 200125
China

Dear Dr. Lei,

Thank you for submitting your revised manuscript entitled "Ccq1-Raf2 interaction mediates CLRC recruitment to establish heterochromatin at telomeres". We would be happy to publish your paper in Life Science Alliance pending final revisions necessary to meet our formatting guidelines.

- please add ORCID ID for the corresponding (and secondary corresponding) author-you should have received instructions on how to do so
- please add the Twitter handle of your host institute/organization as well as your own or/and one of the authors in our system
- please note that titles in the system and manuscript file must match
- please consult our manuscript preparation guidelines <https://www.life-science-alliance.org/manuscript-prep> and make sure your manuscript sections are in the correct order
- please add a callout for Figure S2A to your main manuscript text
- please add scale bars for Figure S2D

LSA now encourages authors to provide a 30-60 second video where the study is briefly explained. We will use these videos on social media to promote the published paper and the presenting author. Corresponding or first-authors are welcome to submit the video. Please submit only one video per manuscript. The video can be emailed to contact@life-science-alliance.org

A. FINAL FILES:

B. MANUSCRIPT ORGANIZATION AND FORMATTING:

Sincerely,

Eric Sawey, PhD
Executive Editor

August 25, 2021

RE: Life Science Alliance Manuscript #LSA-2021-01106-TRR

Prof. Ming Lei
Shanghai Ninth People's Hospital, Shanghai Jiao Tong University School of Medicine
Shanghai Institute of Precision Medicine
115 Jinzun Road
Shanghai, Shanghai 200125
China

Dear Dr. Lei,

Thank you for submitting your Research Article entitled "Ccq1-Raf2 interaction mediates CLRC recruitment to establish heterochromatin at telomeres". It is a pleasure to let you know that your manuscript is now accepted for publication in Life Science Alliance. Congratulations on this interesting work.

DISTRIBUTION OF MATERIALS:

Again, congratulations on a very nice paper. I hope you found the review process to be constructive and are pleased with how the manuscript was handled editorially. We look forward to future exciting submissions from your lab.

Sincerely,
